# A Compact Broadband Power Combiner for High-Power, Continuous-Wave Applications

**DOI:** 10.3390/mi15020207

**Published:** 2024-01-30

**Authors:** Zihan Yang, Qiang Zhang, Kelin Zhou, Lishan Zhao, Jun Zhang

**Affiliations:** 1College of Advanced Interdisciplinary Studies, National University of Defense Technology, Changsha 410073, China; 15234534500@163.com (Z.Y.); zqiang1984@163.com (Q.Z.); zhoukelin98@163.com (K.Z.); 2College of Sciences, National University of Defense Technology, Changsha 410073, China

**Keywords:** power combiner, power divider, RF solid-state source, continuous wave, power synthesis technology

## Abstract

A compact broadband combiner with a high power capacity and a low insertion loss, which is especially useful for solid-state power sources where multi-way power synthesis is needed, was designed and experimentally investigated. The combiner could combine the microwave signals of sixteen terminals into a single one and was based on a radial-line waveguide whose circumferential symmetry benefited the amplitude and phase consistency of the combiner. Simulation and experimental results showed that the prototype device, designed for S-band applications, exhibited a reflection coefficient S_1,1_ < −20 dB in the range of 2.06–2.93 GHz, which corresponds to a relative bandwidth of approximately 34.6%. At 2.45 GHz, the phase imbalance was ±4.5° and the 16-way transmission coefficient was concentrated around −12.0~−12.3 dB. The insertion loss of the device at ambient and elevated temperatures was simulated and experimentally verified, which is of importance for the design of similar high-power microwave combiners. High-power tests proved that even without enforced wind or liquid cooling, the device can handle continuous power (CW) of at least 3.9 kW, which can be much enhanced by taking regular cooling measures. The combined features of the designed combiner suggest promising applications for power synthesis in high-power, solid-state RF sources.

## 1. Introduction

Power synthesis technology is of importance in a wide range of microwave applications, including, for example, microwave heating, wireless communications, microwave plasma equipment, and others [1,2]. In typical solid-state, high-power microwave equipment, the total output power is often offered by multiple power amplifiers (PAs) based on semiconductor RF transistors whose individual output power, due to some fundamental reasons, is insufficient for the desired power level of the whole system [3,4]. Thus, power synthesis is of great importance to obtain high output power from the entire system, and the performance of the combiner, which is the device responsible for RF power synthesis, has a significantf influence on the performance of the whole system.

For high-power, solid-state RF systems where the desired total power is on the order of several to a few tens of kilowatts, making the combiner have a low insertion loss, a wide bandwidth, and compactness is usually a challenging task. Traditional microstrip transmission line combiners, such as Wilkinson combiners, commonly employ a binary tree structure. However, when more input ports are required, they can result in significant losses for multiplex structures [5,6,7,8]. Furthermore, waveguide combiners have garnered significant attention in research due to their favorable characteristics, including low loss, high power capacity, and excellent amplitude and phase consistency.

Recently, German researchers have developed a novel 16-way power combiner for the Ka-band utilizing waveguide transitions. This power combiner exhibits good performance, characterized by an insertion loss (IL) of below 0.24 dB, a return loss of greater than 17 dB, and a phase imbalance of less than 3° across the entire Ka-band [9]. In parallel, Chinese researchers have successfully designed a double-layer wide-band radial-line waveguide power divider/combiner. This design has good bandwidth and achieves a power capacity exceeding 140 MW during pulsed high-power testing [10].

Based on the current literature, there is considerable industry research focusing on passive devices like power dividers and combiners. These devices are primarily designed for low-power civilian microwave signal transmission and military high-power microwave pulse systems. Furthermore, these devices often involve trade-offs—either a compact design with excellent transmission performance but a limited power capacity or a design with a high power capacity at the expense of a larger size and weight. In light of these considerations, we propose a compact wide-band combiner and have conducted innovative research to explore its performance in high-power continuous wave systems [11,12,13].

In this paper, a compact, broadband high-power combiner, which is suitable for high-power CW RF systems with multiple solid-state power amplifiers, is introduced.

The proposed combiner consists of multiple ports for input signals on the periphery of a radial-line waveguide and a coaxial port for output power. The symmetry of the radial-line waveguide permits high amplitude and phase consistency for the input signals. Experimental results demonstrated that the proposed combiner will achieve a reflection coefficient of less than −20 dB within a relative bandwidth of 34.6%. High-power tests conducted at 2.45 GHz revealed that the combiner can accommodate a continuous-wave power capacity of up to 3.9 kW without enforced wind or liquid cooling, which can be greatly enhanced with decent cooling measures in practical applications. The tested insertion loss was as low as 0.21 dB at ambient temperature and was approximately doubled at elevated temperatures of 80 °C. The thermal effects on the transmission performance of the device under high-power, continuous-wave conditions are introduced in detail.

## 2. Design and Simulation

The proposed combiner was designed based on the following ideas: (1) The main body of the device employed a radial-line waveguide, while the output port of the combiner was connected to a coaxial waveguide. (2) To ensure impedance matching across a wide frequency range, a transition section was incorporated within the radial-line waveguide, facilitating the connection between -it and the coaxial waveguide. (3) Sixteen output ports were connected to a special transition-space structure and were evenly distributed on the outside of the radial-line waveguide [14,15].

This design guaranteed that no cascade structure would be needed; thus, the insertion loss could be reduced, the working bandwidth was enlarged by the transition section, and the power capacity was high. The structure of the proposed power combiner is shown in Figure 1.

Ports 2–17 serve as the input ports of the combiner, each having a coaxial waveguide as the feeding line to be connected to the radial-line waveguide, and Port 1 functions as the designated output port for the combiner. Both the outer and the inner radius of the coaxial waveguide connected to port 1 varies gradually as approaching the radial line waveguide. More details are given in the next section. 

### 2.1. Radial Line Waveguide and Transition Section

As shown in Figure 2a, the radius of the radial-line waveguide is *R*, and the distance between the upper and lower metal plates is *H*. The coaxial waveguide’s outer and inner diameters are *R*_1_ and *R*_2_, respectively. The outer conductor of the coaxial waveguide is connected to the feed port of the radial-line waveguide. To achieve single-mode transmission, *H* and *R*_2_ should meet the following conditions:(1)$$H<\frac{\lambda }{2}\;\;\;R_{2}<\frac{\lambda }{2\pi }$$

The impedance between the coaxial waveguide and the radial-line waveguide is not continuous. Therefore, a transition structure needs to be added to lessen the reflection caused by the impedance mismatch. A general impedance matching method is adding an N-layer metallic disc-like transition structure, and the higher the order N of the structure, the better the bandwidth performance [16,17]. When N tends to infinity, the transition structure can be regarded as a continuous curve that is rotated around the *Z*-axis. Therefore, a transition structure is designed, as shown in Figure 2b, which is formed by quarter-elliptic *curve*1 and *curve*2 as generatrix to rotate around the *Z*-axis. An elliptic arc has a higher degree of freedom compared to a circular arc.

This transition structure has the following advantages: (1) Good parameter space for adjustments. *Curve*1 and *Curve*2 are quarter elliptic curves, and the impedance matching can be achieved by optimizing the parameters *A*, *B*, *C*, and *D*; (2) Broadband. The reason is already explained; (3) Uniform electric field distribution. The smooth surface of this structure can effectively avoid breakdown caused by an excessive local electric field and is favorable for improving the power capacity; (4) Good circumferential symmetry. The structure has circumferential symmetry and facilitates the realization of equal amplitude and same-phase power synthesis.

### 2.2. RF Coaxial Connector and Structure at the Input

As depicted in Figure 3a,b, 16 N-type RF coaxial connectors, with a long inner conductor that is inserted into the transition structure of the radial-line waveguide, are connected to the input of the combiner. The radius of the inner and outer conductors of connectors is *R*_3_ = 5 mm, and *R*_4_ = 1.5 mm, respectively, and the dielectric constant of the filling medium is 2.02.

The input of the power combiner is attached to a specially designed structure that creates an air channel with *Face*1 on coaxial connectors and *Face*2 of the radial-line waveguide on the others, as is shown in Figure 4, and this structure also helps to improve the isolation between the input ports. *Face*2 is part of a circular surface whose radius and height are *R* and *H*, respectively. *L* represents the distance between *Face*1 and *Face*2. The symbol *Angle* in Figure 4 represents the arc degree. The arc should meet *Angle* ≤ 360/*n* and *n* is the number of input terminals.

### 2.3. Coaxial—Rectangular Waveguide Mode Converter

The output port in Figure 1, which refers to a coaxial waveguide with a relatively big outer radius, is not convenient for experimental tests or practical applications. Therefore, a coaxial-to-rectangular waveguide mode converter is designed to connect the output of the combiner to a standard WR340 waveguide. The bandwidth of the mode converter needs to be optimized since it may limit the overall bandwidth of the composite structure of the combiner and mode converter. 

There are two ways to improve the bandwidth characteristics of the mode converter. The first is to introduce a ridge waveguide, which permits wide-band characteristics under single-mode transmission conditions since the cutoff wavelength of its basic mode TE_10_ is longer than that in a normal rectangular waveguide, and its field distribution resembles that of the latter. The second is to introduce a circular table into the mode converter as a transition structure in order to achieve impedance matching between the two waveguides. The structural model of the coaxial-to-rectangular waveguide mode converter is shown in Figure 5.

The mode converter uses a coaxial waveguide as the input terminal, whose radius of the inner and outer conductors is identical to that of the coaxial waveguide at the output of the combiner described above. Other parameters of the mode converter are shown in Figure 5b. In addition, efforts are made to release potential electric field concentration by chamfering the corners of the converter (with radius *r*) [18,19].

### 2.4. Simulation

The values of each parameter of the combiner are defined in Table 1, and *Angle* = 16.8°.

The corresponding simulation results of the combiner are shown in Figure 6. *S*_11_ and *S*_21_ are given as representatives and *S*_21_~*S*_17,1_ are the same due to the symmetry of the combiner. The reflection coefficient *S*_11_ is better than −20 dB in the range of 1.97–2.92 GHz (the relative bandwidth is approximately 38.9%). The combiner has good transmission performance at 2.45 GHz, the main targeted frequency. The reflection and transmission coefficients are *S*_11_ = −48.55 dB and *S*_21_ = −12.04 dB, respectively.

## 3. Cold Tests and Discussion

The experimental setup is illustrated in Figure 7. The combiner was tested using a network analyzer with two cables connected to two N-type RF coaxial connectors.

The results of the combiner’s *S* parameters are shown in Figure 8. The reflection coefficient *S*_11_ of the combiner is less than −20 dB in the range of 2.065–2.93 GHz (the relative bandwidth is about 34.6%). The experimental results show that the combiner has a low reflection in broadband and matches the simulation results.

As shown in Figure 8b, the transmission coefficients *S*_n,1_ of 16 input ports are in the range of −11.8 dB to −12.6 dB, and the majority of them are between −12.0 dB and −12.3 dB. A few *S*_n,1_ have a certain difference with the ideal transmission coefficient of −12.04 dB. For example, *S*_11,1_ is approximately −11.8 dB, and *S*_2,1_ is around −12.6 dB. Possible reasons for this include, for example, imperfect symmetry and dimensional deviations from the designed values caused during the processing progress. With that said, such a difference has little impact on the combiner’s overall performance.

Figure 9 shows the phase test results of the 16-way microwave transmission of the combiner. At 2.45 GHz, the phase difference between *S*_4,1_ (29.22°) and *S*_5,1_ (21.14°) is the largest. The phase imbalance of the transmission of the combiner is less than ±4.5°. The results verify that the combiner can deliver high-quality 16-way power synthesis with good amplitude and phase consistency in a broadband range.

In conclusion, the performance of the combiner in the range of 1.6–2.8 GHz has been tested, and the simulation and experimental results show good consistency.

## 4. High Power Tests and Discussion

This section encompasses the experimental evaluation of the combiner’s CW power capacity. Additionally, an assessment is conducted to analyze the impact of device heating effects, under high-power CW conditions on both the power capacity and the transmission performance of the combiner. These investigations aim to validate the effectiveness of the design.

### 4.1. Experimental Setup

Figure 10 illustrates the experimental setup utilized to test the combiner’s power capacity and insertion loss under CW conditions.

The input power *P*1 of combiner 1 and output power *P*2 of combiner 2 can be calculated by the readings of power meters 1 and 2, respectively. Since combiner 1 and combiner 2 are identical, the insertion loss *IL* of a single combiner may be estimated using the following formula: *IL* = (*P*1 − *P*2 − 0.4 dB)/2, where 0.4 dB is the insertion loss of the coaxial cables used in this work.

### 4.2. Experiments on Insertion Loss versus Temperature Variation

After inputting a continuous wave to passive microwave devices, microwave energy loss caused by the insertion loss of the devices is converted into thermal energy, which affects the devices’ transmission performance and power capacity. The impact of thermal effects on insertion loss of the fabricated combiner is examined.

Theoretically, if the input power of the combiner is not high enough to cause thermal runaway, with a fixed input power, the device’s temperature tends to gradually rise until it reaches a stable state. This is verified by experiments whose results are shown in Figure 11, where a continuous RF wave with a power of 2700 W is applied to the combiner.

As depicted in Figure 11, it is clear that as the injection time of the 2700 W continuous wave increases, the temperature at point P rises gradually from 18.2 °C to 86.8 °C. Simultaneously, the insertion loss of the combiner gradually escalates from an initial value of 0.205 dB to 0.43 dB as the temperature increases. It is noted that the increment of insertion loss and that of the device’s temperature is highly consistent, which is further illustrated in Figure 12. This substantiates the detrimental effect of temperature elevation on the combiner’s transmission performance over extended periods of continuous wave injection. Consequently, in addition to benefits on enhancing power capacity and avoiding thermal runaway, implementing active cooling measures such as air or water cooling is of paramount importance to ensure the optimal performance of a power combiner, because it can help keep the temperature of combiner low thereby keeping the insertion loss low.

### 4.3. Experiments on Stable Temperature versus Input Power

For safety considerations and to avoid any damage to the device, we utilized multiphysics field simulation software to simulate the electromagnetic-thermal effect of the combiner when subjected to high-power CW injection.

Figure 13 displays the stabilized temperature of the combiner when subjected to a 3000 W continuous wave through electromagnetic thermal simulation. The combiner’s highest temperature is 95.3°C, located at the inner conductor of the connector at the input port. Meanwhile, the temperature at the point P is 76.7 °C. Conducting thermal effect simulation enables us to estimate the device’s temperature under varying power conditions before employing the experiment.

We conducted an experimental study to examine the thermal effects of injecting CW microwaves with a relatively safe power level (300~4000 W) into the combiner. At each power level, the combiner was subjected to power injection for a long duration (approximately two hours) until the surface temperature at the P-point of the combiner reached its maximum stable value. Subsequently, the temperature was recorded. This approach allowed for a comprehensive analysis of the combiner’s thermal behavior and facilitated the seeking of appropriate operating parameters for optimal performance and device longevity.

The experimental findings, as illustrated in Figure 14, demonstrate a direct correlation between the input power and the stabilization temperature of the P-point on the combiner. With an input power of 340 W, the P-point temperature settles at 34.7 °C, whereas at 3900 W, the temperature rises to 115 °C. These results confirm the combiner’s ability to sustain long-term stability under continuous wave conditions of 3900 W. Note that no dedicated cooling measures were taken in this study for the prototype combiner. We therefore anticipate a much higher upper limit for the device’s power capacity under proper cooling conditions.

These experimental findings serve as validation for the proposed combiner’s ability to achieve high-power, low-loss, multiway power synthesis under continuous wave conditions.

## 5. Conclusions

This article designed a 16:1 power combiner using a radial-line waveguide configuration. The combiner features 16 input ports located outside the radial-line waveguide and the output is connected to a coaxial-WR340 rectangular waveguide mode-converter. The inclusion of transition structures in the radial-line waveguide and a ridge waveguide in the mode converters enhances the combiner’s bandwidth. The symmetry of the structure of the combiner provides good amplitude and phase consistency.

Our simulation validated the feasibility of our designed structure. Cold measurements demonstrate that the combiner achieved a relative bandwidth of 34.6%, with a reflection coefficient of less than −20 dB. The measurements match closely with our simulations. High-power tests indicate that the prototype device can sustain a continuous-wave power capacity of 3.9 kW without internal or external cooling mechanisms. The insertion loss of the combiner at ambient temperature is approximately 0.2 dB. Experimental results confirm that the thermal effects experienced under continuous wave conditions detrimentally impact the device’s transmission performance, highlighting the importance of mitigating thermal effects. Hence, with the implementation of active cooling measures such as internal cooling water pipes or external forced convection, it is anticipated that the device’s power capacity can be greatly increased and the proposed combiner may have important, real applications needing large CW RF power.

## Figures and Tables

**Figure 1 micromachines-15-00207-f001:**
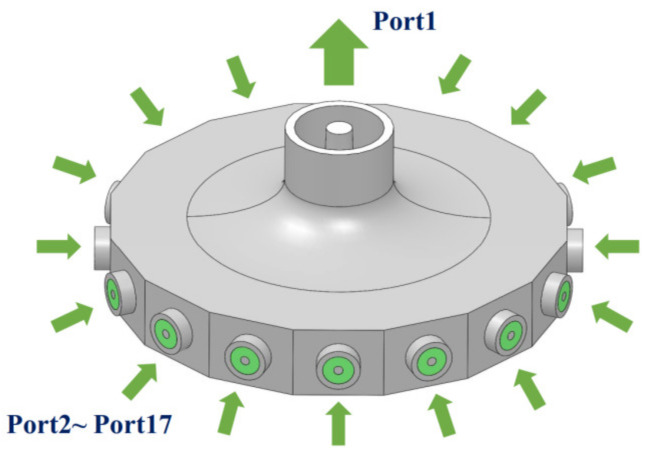
Illustration of the of the proposed power combiner.

**Figure 2 micromachines-15-00207-f002:**
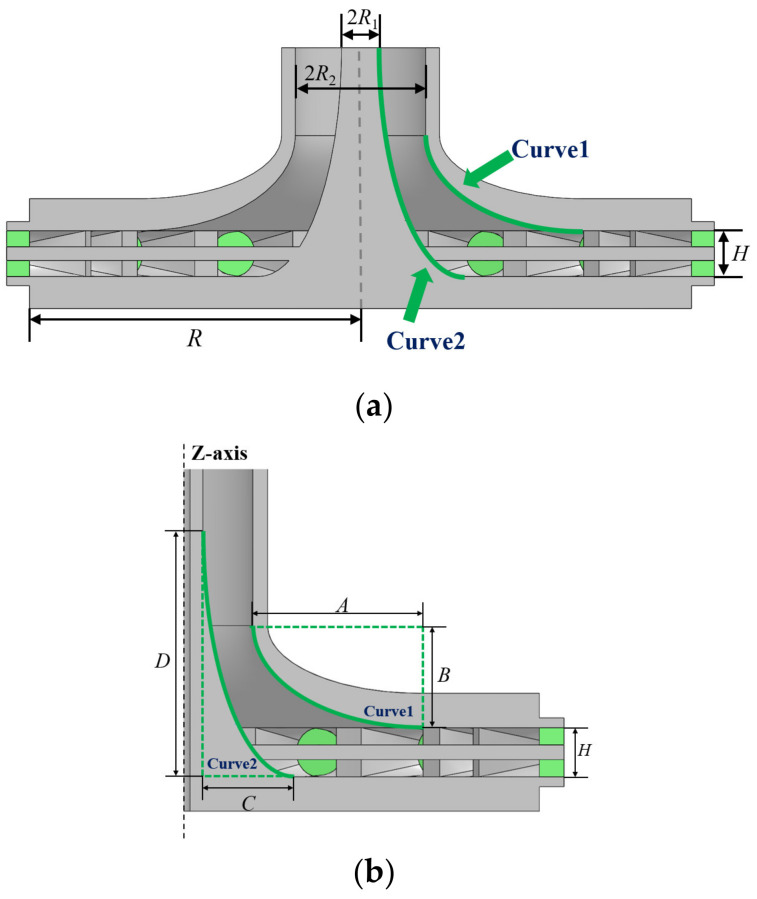
Profile view of the power combiner: (**a**) overall sectional structure and (**b**) transition structure of the power combiner. The green area represents the medium within the joint, while the gray area represents the metal. (**a**) displays the parameters for the radial line waveguide and coaxial waveguide in the proposed combiner and the transition structure connecting them. (**b**) provides a more detailed view of this transition structure and its parameters.

**Figure 3 micromachines-15-00207-f003:**
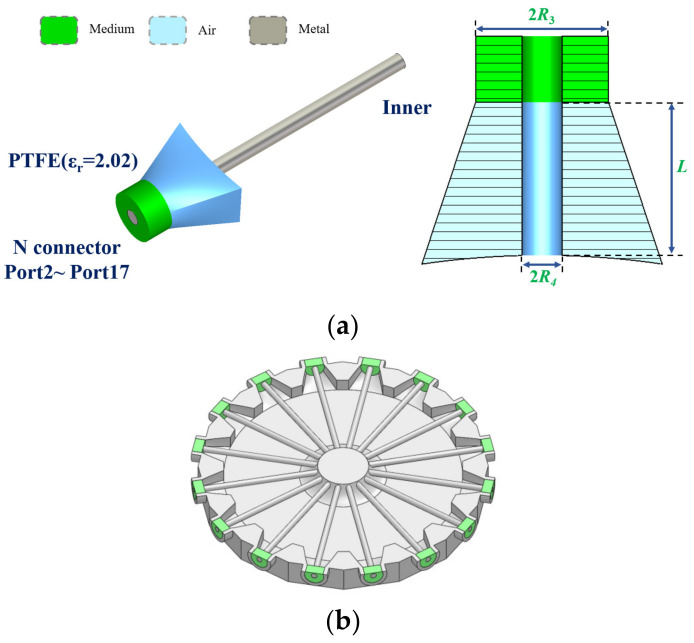
Structure of the input side: (**a**) N-type RF coaxial connector and (**b**) profile view. (The joint’s medium is represented in green, the internal channel of the combiner (also the electromagnetic wave transmission area) in blue, and the metal region in gray. (**a**) shows an input connector model, while (**b**) is a cross-section of all input connectors within the combiner).

**Figure 4 micromachines-15-00207-f004:**
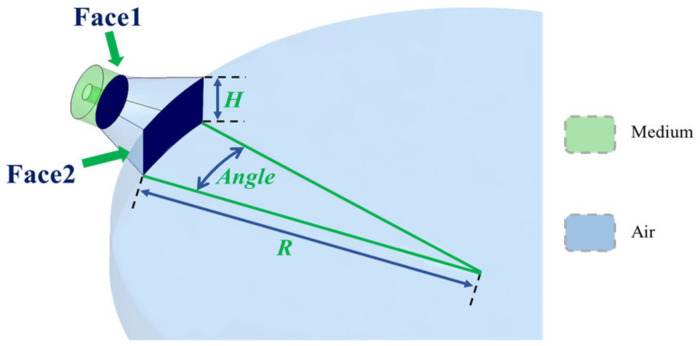
Profile view of the input side. This diagram shows the transitional internal space channel at one of the combiner’s inputs and how it connects to the radial line waveguide. Other input ports on the combiner are evenly distributed around the outside of the waveguide in this particular model.

**Figure 5 micromachines-15-00207-f005:**
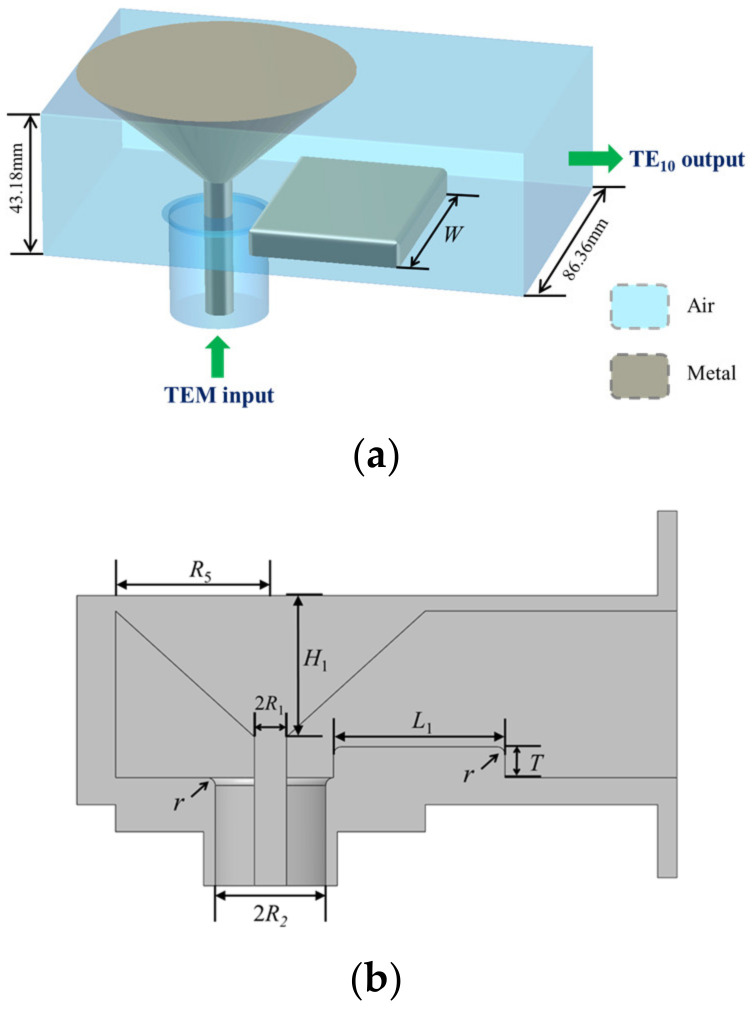
Structure of the model converter: (**a**) perspective view and (**b**) profile view. The proposed mode converter is shown in (**a**), with the blue section representing the internal spatial channel for electromagnetic wave transmission and the gray area describing the metal. (**b**) shows a cross-sectional view of the mode converter, including all structural parameters.

**Figure 6 micromachines-15-00207-f006:**
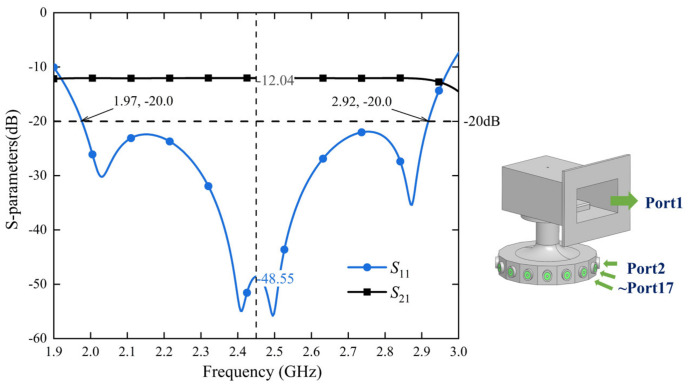
Results of *S*-parameter simulation of the power combiner. The figure shows the frequency on the horizontal axis and the *S*-parameter simulation result of the combiner on the vertical axis. For the 1:16 power divider/combiner simulation, the ideal single-way transmission coefficient *S*_n,1_ is approximately −12.04 dB, calculated as 10 log (1/16). During simulation, when Port1’s reflection coefficient *S*_11_ is better than −20 dB, transmission coefficient *S*_21_~*S*_17,1_ will be around −12.04 dB due to the combiner’s excellent symmetry, while good power distribution/synthesis can be realized.

**Figure 7 micromachines-15-00207-f007:**
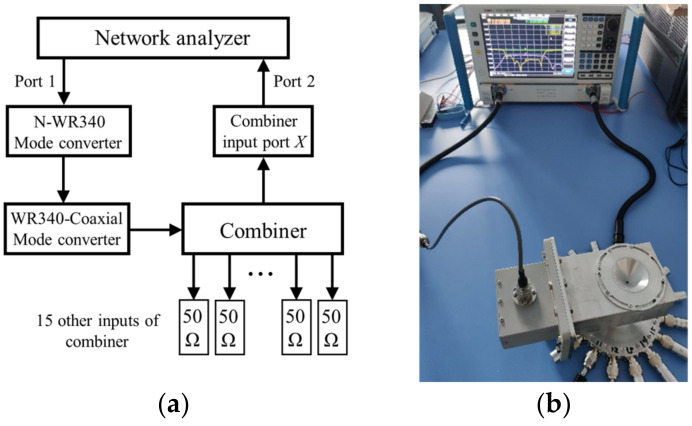
Cold tests: (**a**) the block diagram and (**b**) a snapshot of the experimental setup. The output port of the combiner is connected to a mode converter (WR340~N-type RF coaxial) and then to port 1 of the network analyzer. Fifteen output ports of the combiner are connected to the standard 50 Ω matched load, and the remaining port *X* to be tested is connected to the network analyzer port 2. The transmission coefficients and phases of the 16 input ports are measured sequentially in the experiment.

**Figure 8 micromachines-15-00207-f008:**
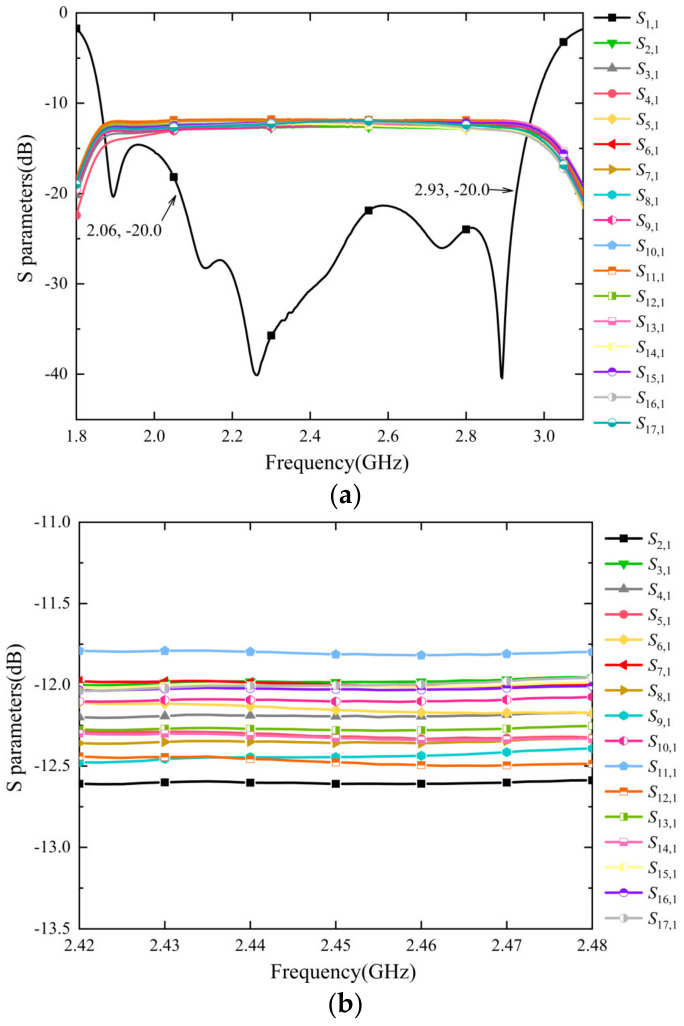
Results of *S*-parameter measurements: (**a**) *S*-parameter measurement results within a wide frequency band range and (**b**) transmission coefficients measured. It is seen from (**a**) that the reflection *S*_11_ is less than −20 dB in the range of 2.065–2.93 GHz. (**b**) shows the measured transmission coefficient *S_n_*_,1_ from port 2~17 to port 1. The transmission coefficients of the 16 ports are distributed within a narrow range around −12 dB.

**Figure 9 micromachines-15-00207-f009:**
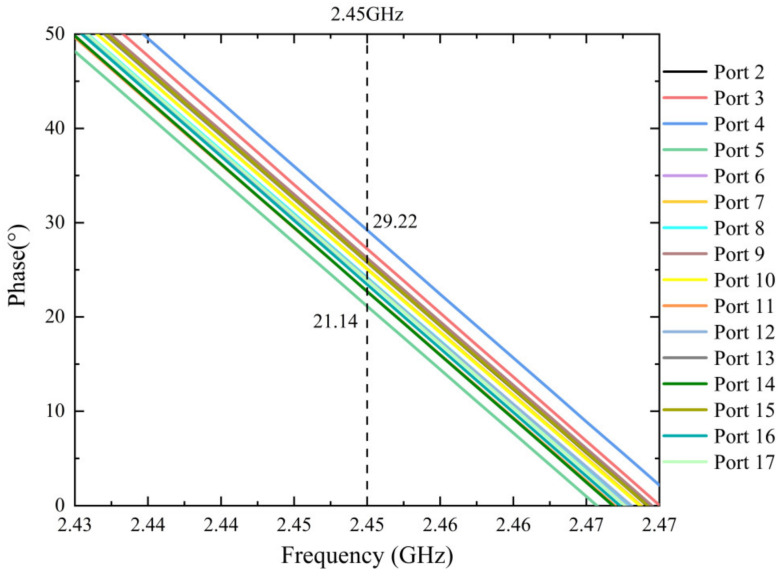
Phase measured (the figure shows the measured transmission phase from port 2~17 to port 1).

**Figure 10 micromachines-15-00207-f010:**
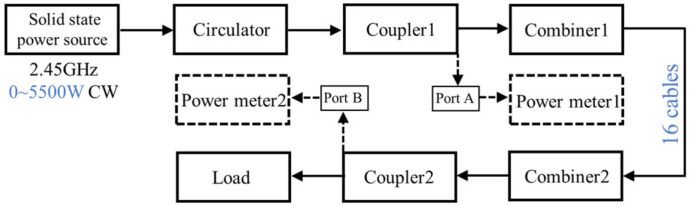
Block diagram of the experimental setup. A high-power CW RF source is connected to combiner 1 and combiner 2. The former here is used as a power divider, to divide the input microwave signal into 16 ways, which are subsequently connected to the inputs of combiner 2 through coaxial cables. Power meters 1 and 2 are attached to the coupling Port A of coupler 1 and Port B of coupler 2, respectively, to measure the coupled input and output power (the microwave energy is relatively low and falls within the power measurement range of the power meter). The load is employed to absorb the high-power microwave energy output from the straight-through end of coupler 2, thereby preventing any potential damage to other devices. In addition, a temperature sensor is connected to the combiner’s outer surface to record the device’s temperature.

**Figure 11 micromachines-15-00207-f011:**
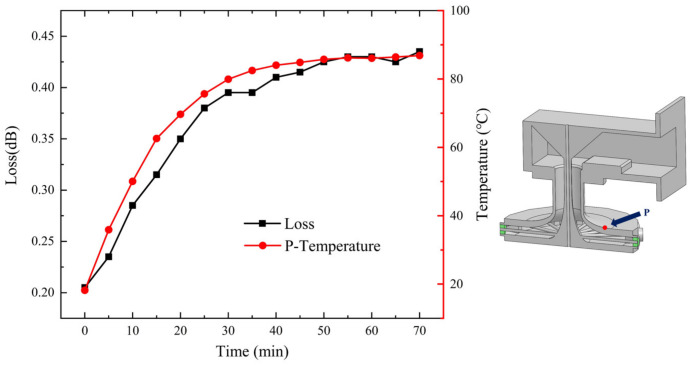
The insertion loss and P-point temperature of the combiner vs. time, for a power of 2700 W in the power combiner. The horizontal axis represents the time of power injection to the combiner. The left vertical axis indicates the insertion loss of the device, and the right vertical axis is the temperature at point P indicated in the right panel.

**Figure 12 micromachines-15-00207-f012:**
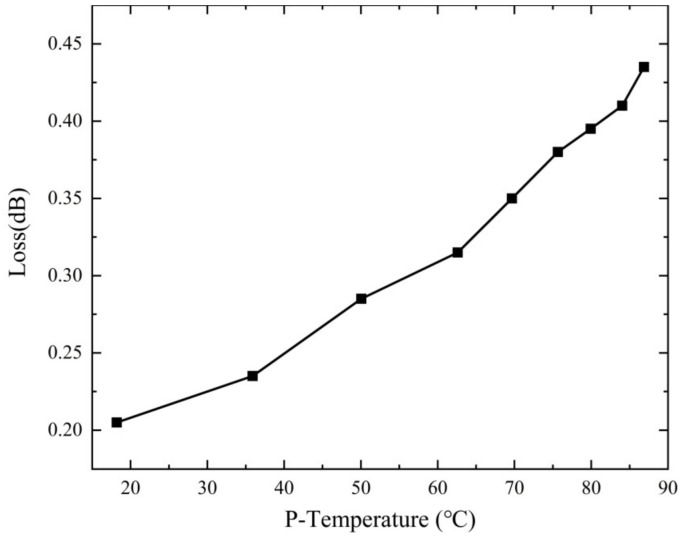
Relationship between insertion loss and temperature of the device (this graph shows how the insertion loss of the combiner changes with the temperature; the horizontal axis represents the temperature, while the vertical axis represents the insertion loss).

**Figure 13 micromachines-15-00207-f013:**
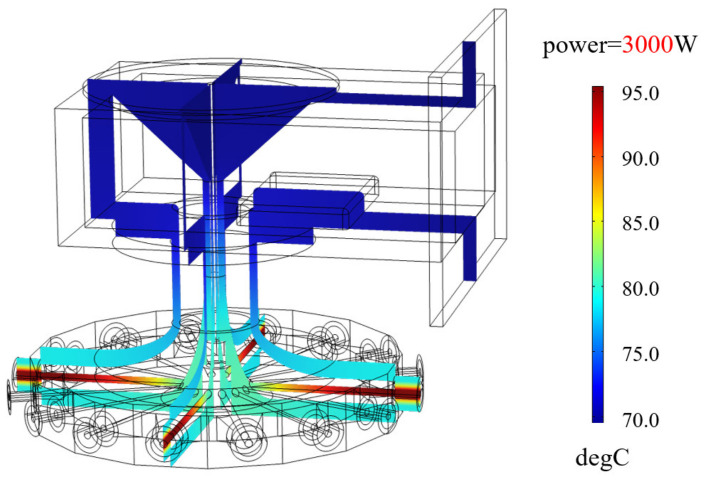
Simulated steady-state temperature distribution of the combiner subjected to 3000 W CW injection at 2.45 GHz. In the simulation process, for the purpose of enhancing the stringency of simulation conditions, the convective heat transfer coefficient between the metal and air was adjusted to 5 W/(m^2^·K), and the thermal conductivity of the 6061 aluminum alloy was configured to 155 W/(m^2^·K). The displayed results were obtained with electromagnetic and thermal multiphysics field simulation. Since in experiments of high-power operation, it was observed that the inner conductor of the input port connector exhibited the highest temperature, while the outer surface of the waveguide showed the lowest temperature, which is consistent with the simulated results above, one can believe that by measuring the temperature at point P, it is possible to estimate the highest temperature point of the combiner.

**Figure 14 micromachines-15-00207-f014:**
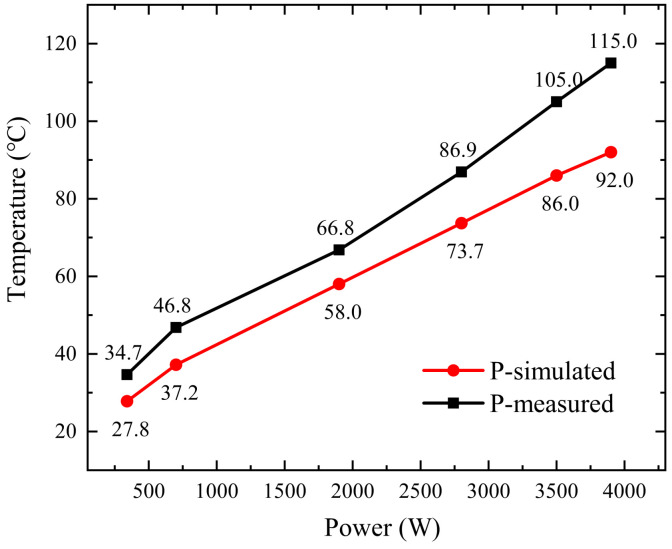
The P-point temperature of the combiner vs. input power. The stabilized temperatures at point P of the combiner under various power conditions are shown in the figure. It is seen that the stabilized temperature increases monotonically with respect to the input power.

**Table 1 micromachines-15-00207-t001:** Structure parameter of power combiner.

Parameters	*R*	*H*	*R* _1_	*R* _2_	*A*	*B*	*C*	*D*
Values/mm	60.8	10.0	4.1	14.3	33.9	20.8	18.1	50.0
Parameters	*L*	*R* _5_	*H* _1_	*L* _1_	*W*	*T*	*r*	
Values/mm	11.6	40.0	33.0	44.3	63.0	9.4	2	

## Data Availability

Data are contained within the article.

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
