# Peer review of "A Compact Broadband Power Combiner for High-Power, Continuous-Wave Applications"

_micromachines, 2024, doi:10.3390/mi15020207_

Round 1

Reviewer 1 Report

Comments and Suggestions for Authors

The paper proposes a CW power combiner with simulation results closely matching experimental results for insertion loss and high-power capacity, etc. The following points can be considered to improve the paper:

1)    The paper needs a proofread. For instance, the following paragraph:

Traditional combiners based on 37 micro-strip transmission lines (e.g. Wilkinson combiners) typically have a one-in-two tree 38 structure and are generally cascaded to obtain high power output, and such cascade struc- 39 ture harms the efficiency of power synthesis, and in addition, the power capacity is also 40 limited by the and micro-strip transmission lines. Heat dissipation is often a an issue in 41 such structures [5–8]. I 

2)    The introduction of the paper lacks rigor and must be improved using the latest published works on power combiners.

3)    Comparison of proposed CW power combiner with the existing CW power combiners is missing, which must be included to underscore the contribution of the paper.

4)    Figure 10 needs to be better explained in relation to load and power meters being used.

5)    Figure 11 needs to incorporate label for input power on horizontal axis along with time.

Comments on the Quality of English Language

Adequate. However, paper needs a proofread.

Author Response

Dear Reviewer

Thank you for your letter and the constructive comments on this article in your busy schedule. All of us authors have carefully read the comments that you have given us,  and have discussed and revised each of these issues. The following is my list of revisions. In addition, we have resubmitted a new manuscript in the revised state, with the revisions highlighted in red. If there are any incorrect answers or questions in the manuscript please do not hesitate to let us know.

  • The paper needs a proofread. For instance, the following paragraph:

Traditional combiners based on micro-strip transmission lines (e.g. Wilkinson combiners) typically have a one-in-two tree structure and are generally cascaded to obtain high power output, and such cascade structure harms the efficiency of power synthesis, and in addition, the power capacity is also limited by the and micro-strip transmission lines. Heat dissipation is often a an issue in such structures.

Respond:

We tried our best to improve the manuscript and made some changes to the manuscript. These changes will not influence the content and framework of the paper. And here we did not list the changes but marked in red in the revised paper. We appreciate for Editors/Reviewers' warm work earnestly and hope that the correction will meet with approval.

  • The introduction of the paper lacks rigor and must be improved using the latest published works on power combiners.

Respond:

We sincerely appreciate your valuable feedback. We have carefully reviewed the literature and included the latest research status on power combiners in the revised manuscript's introduction section 

  • Comparison of proposed CW power combiner with the existing CW power combiners is missing, which must be included to underscore the contribution of the paper.

Respond:

Thank you for your valuable feedback. Regarding this issue, we have found that existing research on power combiners primarily focuses on bandwidth, insertion loss, and phase imbalance as the main parameters, with limited studies on power capacity. Most of the research on power capacity is based on simulations or experimental testing of pulse power.  Furturemore, limited research has been conducted on the power capacity and performance of combiners under high-power continuous wave. Therefore, we did not include a comparison of the proposed continuous wave power combiner with existing continuous wave power combiners in the manuscript.

Additionally, we have provided relevant explanations in the introduction to highlight the necessity of our research. Once again, we appreciate your valuable feedback.

  • Figure 10 needs to be better explained in relation to load and power meters being used.

Respond:

Thank you very much for your feedback.First, allow me to explain the relationship between the power meter and the load. The purpose of power meter 2 is to measure the microwave energy output from coupled end of coupler 2 (which is a very small amount of energy and will not damage the power meter), in order to estimate the power output from the combiner. The purpose of the load is primarily to absorb the power output from the through end of the coupler (which is a significant amount of power that needs to be absorbed by the load to prevent irreversible damage to other equipment). I apologize for not clarifying this in the text. Therefore, in the revised manuscript, I have added the necessary information and made the appropriate markings.

  • Figure 11 needs to incorporate label for input power on horizontal axis along with time.

Respond:

Thank you very much for your valuable feedback. Please accept my apologies for the oversight in not providing a clear explanation of Figure 11. The actual interpretation of this figure is the relationship between the insertion loss and the surface temperature at point P of the combiner under the continuous injection of 2700W microwave power. I have revised the description of Figure 11 in the manuscript to address this, and I greatly appreciate your assistance in helping me identify and rectify this error in a timely manner.

Thank you for your time and valuable insights.

Also, I will use this opportunity and wish you a happy New Year!

Sincerely,

Zihan Yang

Reviewer 2 Report

Comments and Suggestions for Authors

The work is complete and well described. I just wanted to ask what are your explanations for the difference between the results of the simulations and the cold measurements carried out on the prototype?

Author Response

Dear Reviewer

Thank you for your thorough review of our work and for providing valuable feedback. We acknowledge the differences you pointed out between the simulation results and the actual measurements, and we sincerely apologize for not providing an explanation for this issue in the text. The following is my revision. If there are any incorrect answers or questions in the manuscript, please do not hesitate to let us know.

1) The work is complete and well described. I just wanted to ask what are your explanations for the difference between the results of the simulations and the cold measurements carried out on the prototype?

Respond

Comparing the measured and simulated results, it is evident that there is a certain difference between the two. The primary cause of this discrepancy is attributed to minor mechanical fabrication errors and gaps at the connection points between the conductors within the connectors at the 16 output ports and the transitional structure inside the radial line waveguide. These errors are challenging to simulate accurately in the simulation. However, overall, the measured and simulated results align well, and the experimental outcomes affirm the feasibility of the design. We will provide an explanation in the manuscript regarding the differences between the measured and simulated results. Thank you very much for your valuable suggestions.

Thank you for your time and valuable insights.

Sincerely,

Zihan Yang

Reviewer 3 Report

Comments and Suggestions for Authors

This research has all relevant scientific steps: idea - novel methodology, realization  with simulation and implementation laboratory prototype. All steps are adequate presented and verified. 

The introduction has adequate background with relevant references.  

The novel design is appropriate with presented methodology and realization with simulations.

The experimental results verified all design steps and validated the relevance of novel methodology. 

The conclusions supported by the presented results, but authors have to add one new part with discussion in the context of comparison between this novel realization and realizations given in the reference list. 

The authors should check template for references and correct all references.

Author Response

Dear Reviewer

Thank you for your letter and the constructive comments on this article in your busy schedule. All of us authors have carefully read the comments that you have given us,  and have discussed and revised each of these issues. The following is my list of revisions. In addition, we have resubmitted a new manuscript in the revised state, with the revisions highlighted in red. If there are any incorrect answers or questions in the manuscript please do not hesitate to let us know.

  • The conclusions supported by the presented results, but authors have to add one new part with discussion in the context of comparison between this novel realization and realizations given in the reference list.

Respond:

We sincerely appreciate your valuable feedback. Related additions we've made in the introduction

  • The authors should check template for references and correct all references.

Respond:

Thank you for your valuable feedback. We have compared the proposed combiner with existing counterparts and provided additional explanations in the introduction section.

Thank you for your time and valuable insights.

Also, I will use this opportunity and wish you a happy New Year!

Sincerely,

Zihan Yang

Round 2

Reviewer 1 Report

Comments and Suggestions for Authors

The authors have addressed my comments.